# Coreset Selection for Object Detection

## Abstract

Coreset selection is a method for selecting a small, representative subset of an entire dataset. It has been primarily researched in image classification, assuming there is only one object per image. However, coreset selection for object detection is more challenging as an image can contain multiple objects. As a result, much research has yet to be done on this topic. Therefore, we introduce a new approach, *Coreset Selection for Object Detection* (CSOD). CSOD generates imagewise and classwise representative feature vectors for multiple objects of the same class within each image. Subsequently, we adopt submodular optimization for considering both representativeness and diversity and utilize the representative vectors in the submodular optimization process to select a subset. When we evaluated our method on the Pascal VOC dataset, our method outperformed random selection by +6.8%p in $AP_{50}$ when selecting only 200 images.

## 1 Introduction

Driven by the exponential growth of data volume and model sizes in various domains such as image classification (Radford et al., 2021), object detection (Zong et al., 2022), and image segmentation (Cheng et al., 2022), deep learning has witnessed remarkable advancements. However, these advancements have also raised critical challenges related to computational costs and memory constraints, mainly due to the numerous experiments involved in tasks such as model search and hyperparameter tuning. Coreset selection is a method to select the most informative data points within a labeled dataset, aiming to maintain the model's performance without using the entire labeled dataset.

While coreset selection also has been extensively researched in the context of active learning (Sener & Savarese, 2017), where it is used to select valuable unlabeled data, in this paper, we will focus on its application to labeled data, aiming to identify the most informative subset of the labeled dataset for supervised learning. We aim to select a subset, or "coreset," that encapsulates the essential features or distribution of the entire dataset, enabling efficient training and analysis.

The basic idea is to identify the "coreset" that captures the essential features or distribution of the entire dataset. This coreset can then be used for training or analysis instead of the complete data, making the process more efficient regarding computational resources and time. Coreset selection can also be applied in various contexts, including neural architecture search (Shim et al., 2021) and continual learning (Borsos et al., 2020). However, it has primarily been studied extensively in image classification with one-hot labels. In this paper, we extend the concept of coreset selection to the object detection task and propose a new method for coreset selection in object detection.

Let us consider selecting the most informative coreset among the labeled dataset for classification tasks. Various coreset selection methods have been designed based on different criteria. For

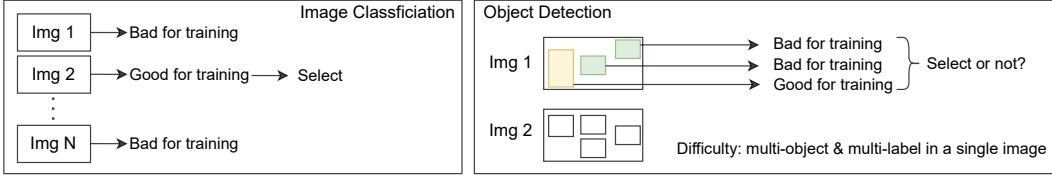

Figure 1: The difference in coreset selection between image classification and object detection.

example, Braverman et al. (2019) extracted feature maps from images and performed clustering, while Coleman et al. (2019) proposed an uncertainty-based approach using confidence scores for image classification. Guo et al. (2022) demonstrated that utilizing submodular functions (Krause & Golovin, 2014) is effective for coreset selection in image classification. However, as depicted in the left of Figure 1, coreset selection has been predominantly explored for image classification data, and their specific approaches were based on the assumption that there is only one object per image.

On the contrary, the right of Figure 1 illustrates the challenges of coreset selection in object detection compared to image classification. In object detection, a single image can contain multiple objects. If we evaluate the suitability of an image on an object-by-object basis, considering one object as suitable for learning does not necessarily imply that the others are also suitable. In other words, the determination of a core image should be based on all objects in an image.

Therefore, we propose a method that takes the aforementioned challenge into account. We create a class representative vector for an image by averaging the features of objects in the same class within the image, resulting in *imagewise-classwise representative vectors*. These representative vectors can encapsulate multiple objects of the same class within that image.

Furthermore, we employ a *greedy approach* to select individual data points sequentially by class order, gradually constructing the coreset. This method considers only one class at each selection step. Although this method does not consider all objects for every class in an image simultaneously, it ensures that each class's most appropriate selections are made at each selection step.

To ensure that the selected subset informatively represents the entire dataset in terms of both representativeness (how typical the data is) and diversity (how diverse it is), we introduced a submodular function (Krause & Golovin, 2014), which guides us in selecting the most informative subset from the imagewise and classwise average features for each class. As a result, CSOD achieved +6.8%p improvement in $AP_{50}$ over random sampling on Pascal VOC (Everingham et al., 2007) when selecting only 200 images.

In summary, we extend coreset selection into object detection, simultaneously considering multiple objects and localization. While we focus on multi-objects of the same class, future research should explore coreset selection methods that account for objects from different classes in a single image. Our work represents the pioneering effort in coreset selection for object detection, with potential applications in tasks requiring multi-object aspects and localization consideration.

## 2    BACKGROUND AND PRIOR WORKS

### 2.1    CORESET SELECTION

Welling (2009) introduced the concept of herding for iterative data point selection near class centers. Wei et al. (2015) applied submodular functions to the Naive Bayes and Nearest Neighbor classifier. We also adopt the submodular functions, so we provide further explanation in Section 2.3. Braverman et al. (2019); Huang et al. (2019) modified statistical clustering algorithms like k-median and k-means to identify data points that effectively represent the dataset. Coleman et al. (2019) utilized uncertainties measured by entropy or confidence. Huang et al. (2023) theoretically explained the upper and lower bounds on the coreset size for k-median clustering in low-dimensional spaces. However, most previous researches focused on image classification, and to the best of our knowledge, our work is the first research to design coreset selection specifically for object detection.

### 2.2    DATASET DISTILLATION

Dataset Distillation (Wang et al., 2018; Nguyen et al., 2021; Cazenavette et al., 2022; Zhou et al., 2022) focuses on generating synthetic data to capture the overall information of the dataset. While Coreset Selection focuses on selecting the most informative data points from the existing dataset, such synthetic data not only accelerates model training but also holds potential for applications in domains where data privacy is crucial (Dong et al., 2022), as it represents the original data in a transformed form. However, previous Dataset Distillation research mainly focused on image classification. We anticipate that if coreset selection research for object detection is further developed, it will facilitate future studies in dataset distillation designed explicitly for object detection.

Figure 2: The forward process during the training phase of Faster R-CNN. The RoI features include both foreground and background regions at the forward process.

### 2.3 SUBMODULAR FUNCTION

A set function $f : 2^{\mathbb{V}} \to \mathbb{R}$ is considered submodular if, for any subsets $\mathbb{A}$ and $\mathbb{B}$ of $\mathbb{V}$ where $\mathbb{A} \subseteq \mathbb{B}$ and $x$ is an element not in $\mathbb{B}$, the following inequality holds:

$$f(\mathbb{A} \cup \{x\}) - f(\mathbb{A}) \geq f(\mathbb{B} \cup \{x\}) - f(\mathbb{B}) \tag{1}$$

Here, $\Delta(x|\mathbb{A}) := f(\mathbb{A} \cup \{x\}) - f(\mathbb{A})$ represents the benefit of adding $x$ to the set $\mathbb{A}$. In simple terms, this inequality means that adding $x$ to $\mathbb{B}$ provides less additional benefit than adding $x$ to $\mathbb{A}$. This is because $\mathbb{B}$ already contains some of the information that $x$ can offer to $\mathbb{A}$. Therefore, we can use submodularity to find a subset that maximizes the benefit of adding each element.

However, in general, selecting a finite subset $\mathbb{S}$ with the maximum benefit is a computationally challenging problem (NP-hard) (Krause & Golovin, 2014). To address this, we employ a greedy algorithm that starts with an empty set and adds one element at a time. Specifically, $\mathbb{S}_i$ is updated as $\mathbb{S}_{i-1} \cup \operatorname*{argmax}_{x} \Delta(x|\mathbb{S}_{i-1})$. For more information, please refer to Krause & Golovin (2014).

### 2.4 FASTER R-CNN

Various multi-object detectors exist, including Faster R-CNN (Ren et al., 2015), SSD (Fu et al., 2017), YOLO (Redmon & Farhadi, 2018), and DETR (Carion et al., 2020). We selected Faster R-CNN as our base model. This choice was motivated by its widespread adoption not only in supervised detection but also in various research areas such as few-shot detection (Wang et al., 2020), continual learning Wang et al. (2021), and semi-supervised object detection (Jeong et al., 2019).

Faster R-CNN operates as a two-stage detector. As illustrated in Figure 2, the first stage employs Region Proposal Network (RPN) to generate class-agnostic object candidate regions in the image, followed by pooling these regions to obtain Region of Interest (RoI) feature vectors. In the second stage, the model utilizes these RoI feature vectors for final class prediction and bounding box regression. Our research uses these RoI feature vectors for coreset selection.

### 2.5 ACTIVE LEARNING FOR OBJECT DETECTION

Active learning is concerned with selecting which unlabeled data to annotate and is thus related to coreset selection. In the context of active learning for object detection, Yuan et al. (2021) proposed a method based on uncertainty that utilizes confidence scores on unlabeled data. Kothawade et al. (2022) aimed to address the low performance issue in rare classes when conducting active learning. The method extracted features of rare classes from labeled data and aimed to maximize the information of rare classes by submodular function and computing the cosine similarity between these labeled features and the features of unlabeled data.

### 3 PROBLEM SETUP

We have an entire training dataset $\mathbb{T} = \{x_i, y_i\}_{i=1}^{D}$. Here, $x_i \in \mathbb{X}$ is an input image, and $y_i \in \mathbb{Y}$ is a ground truth. Because these data are for object detection, $y_i = \{c_j, b_j\}_{j=1}^{G}$ contains variable numbers of annotations depending on the image. In the $G$ annotations, $c_j$ is a class index, and $b_j = \{b_j^{left}, b_j^{top}, b_j^{right}, b_j^{bottom}\}$ denotes the coordinates of the $j$-th bounding box. Coreset selection aims to choose a subset $\mathbb{S} \subset \mathbb{T}$ that best approximates the performance of a model trained on the entire dataset, $\mathbb{T}$.

In our approach, we prioritize the number of images over the number of annotations. This is because annotations typically consist of relatively few strings, and what primarily affects training time and data storage is the number of images rather than the number of annotations.

## 4 METHOD

The CSOD process can be broadly divided into two steps. First, we extract RoI feature vectors from the ground truth of the entire training image set (not from the output of RPN) (Section 4.1). We average the RoI features of the same class within one image (Section 4.2).

Second, we utilize the averaged RoI feature vectors to greedily select images one by one for each class in a rotating manner (Section 4.3). We introduced submodular optimization to ensure that the selection process considers both representativeness and diversity. Algorithm 1 provides the pseudocode, while additional figures to aid understanding are available in Appendix Figure 7.

---

**Algorithm 1** Pseudocode

---

**Require:** Training Data $\mathbb{T}$ with $C$ classes. Trained backbone $f_\theta$. Roi pooler $g$.
**Ensure:** Selected subset $\mathbb{S}$ with size $N$
1: **for** all data $(x_i, y_i) \in \mathbb{T}$ **do**                              ▷ Stage1
2:     $\mathbb{R}_i \leftarrow$ extract roi feature vectors of $(x_i, y_i)$ by $f_\theta$ and $g$        ▷ Section 4.1
3:     **for** all class $c$ in $y_i$ **do**
4:         $\boldsymbol{p}_{i,c} \leftarrow$ average RoI features of the corresponding class $c$ in $\mathbb{R}_i$    ▷ Section 4.2
5:     **end for**
6: **end for**
7: $\{\boldsymbol{a}_c\} \leftarrow$ empty lists for all class c to store selected features
8: **while** $|\mathbb{S}| < N$ **do**                              ▷ Stage 2
9:     **for** $k \leftarrow 1$ to $C$ **do**                          ▷ Section 4.3
10:         compute score $s_{i,k}$ for all $i$, using the unselected $\boldsymbol{p}_{i,k}$ with previous selected $\boldsymbol{a}_k$  ▷ Eq. 4
11:         select a datum by $\arg\max_i s_{i,k}$ and move the corresponding $\boldsymbol{p}_{i,k}$ into $\boldsymbol{a}_k$
12:     **end for**
13: **end while**

---

### 4.1 GROUND TRUTH ROI FEATURE EXTRACTRATION

With Faster R-CNN, we extract RoI feature vectors from training images by the ground truth (not from the output of the RPN). If the $i$-th training image contains $G$ ground truth objects, then we have $G$ RoI feature vectors, $\mathbb{R}_i$, as follows:

$$\mathbb{R}_i = \{\boldsymbol{r}_{i,j}\}_{j=1}^G = GAP(g(f_\theta(x_i), y_i)), \tag{2}$$

where $x_i$ is an input image, $y_i$ is a ground truth, $f_\theta$ is the backbone trained by the entire data, $g$ is the RoI pooler, $GAP$ is global average pooling and $\boldsymbol{r}_{i,j}$ is the $j$-th RoI feature vector of the $i$-th image.

### 4.2 IMAGEWISE AND CLASSWISE AVERAGE

Once we have extracted all the RoI feature vectors for each image, we have a choice to make: For coreset selection, should we average the RoI feature vectors of the same class within a single image to create a single prototype vector representing that class for the image, or should we use these RoI feature vectors directly?

As mentioned in Section 1, we chose the averaging approach. If $\mathbb{R}_i = \{\boldsymbol{r}_{i,j}\}_{j=1}^G$ represents the RoI feature vectors for the $i$-th data with $G$ ground truth objects, then the average RoI feature vector for class $c$ in the $i$-th data, denoted as $\boldsymbol{p}_{i,c}$, is calculated as follows:

$$\boldsymbol{p}_{i,c} = \frac{1}{|\{j|c_j = c\}|} \sum_{\{j|c_j = c\}} \boldsymbol{r}_{i,j}. \tag{3}$$

## 4.3 GREEDY SELECTION

After obtaining averaged RoI feature vectors, our selection process follows a greedy approach, where we iteratively choose one data point from each class at a time. To facilitate this, we compute a similarity-based score for each RoI feature vector. This scoring mechanism based on submodular function assigns higher scores to RoI feature vectors that are similar to others within the same class and lower scores to those similar to RoI feature vectors that have already been selected. This strategy enables us to take into account previously selected data points when making new selections.

The score function, which is inspired by Iyer et al. (2021), computes the score $s$ for the $i$-th data point within class $k$ as follows:

$$s_{i,k} = \lambda \cdot \sum_j cossim(\boldsymbol{p}_{i,k}, \boldsymbol{p}_{j,k}) - \sum_j cossim(\boldsymbol{p}_{i,k}, \boldsymbol{a}_{j,k}) \qquad (4)$$

The term "$cossim$" represents the cosine similarity, $\boldsymbol{p}_i$ represents the averaged RoI feature vectors that have not been selected yet, and $\boldsymbol{a}_i$ denotes the previously selected RoI feature vectors. The hyperparameter $\lambda$ is introduced to balance the contributions within the scoring function, in which the former term aims to select the most representative one from among those that have not been selected, while the latter term aims to select something different from what has already been selected before. The experiment related to $\lambda$ can be found in Section 5.3.2.

## 5 EXPERIMENTS

In this section, we will empirically demonstrate the effectiveness of CSOD through a series of experiments. Firstly, we will establish that our method outperforms various random selections. We will then investigate the tendency associated with the number of selected images and the hyperparameter $\lambda$ of Eq. (4). Additionally, we will illustrate the performance differences between averaging the RoI features of a class within an image or using the individual features as they are. Furthermore, we will extend our analysis to evaluate the performance on different datasets and various network architectures.

### 5.1 IMPLEMENTATION DETAILS

We conducted experiments on Pascal VOC 2007+2012 (Everingham et al., 2007). We utilized the trainval set during selection and training, and evaluation was performed on the VOC07 test data. The evaluation metric is Average Precision at IoU 0.5 ($AP_{50}$). For ablation and analysis purposes, given that VOC consists of 20 classes, we selected 200 images, trained for 1000 iterations, and set $\lambda$=0.05 in Eq. (4). Due to the limited number of images, we computed the average performance over 20 runs to account for performance variation.

We employed Faster R-CNN-C4 (Ren et al., 2015) with ResNet50 (He et al., 2016). For the selection phase, we used the model weight trained on VOC07+12, provided by detectron2 (Wu et al., 2019). After selection, for the subsequent training by the selected subset, we employed the backbone pre-trained on ImageNet (Deng et al., 2009). Please refer to Appendix A for more detailed experimental settings and Faster R-CNN hyperparameters.

### 5.2 COMPARISON WITH RANDOM SELECTIONS

Figure 3-Left demonstrates that our approach consistently outperforms other selection methods when selecting 200 images. Notably, in this comparison, "# max" and "Ours" are the only methods without randomness, while the rest incorporate some degree of randomness. Therefore, we did not specifically address the performance variance of each selection method. Ours was conducted with a fixed set and without randomness, which results in lower performance variance. This experiment's significance lies in the clearly higher performance of ours compared with those of other methods.

We categorized the random selection into several approaches. "Full random" involves randomly selecting 200 images from the entire dataset, but if there were classes with no objects among the 200 images, we repeated the selection process. "Uniform" and "Ratio" approaches involve sampling images one by one for each class until 200 images are selected (sampling without replacement). In

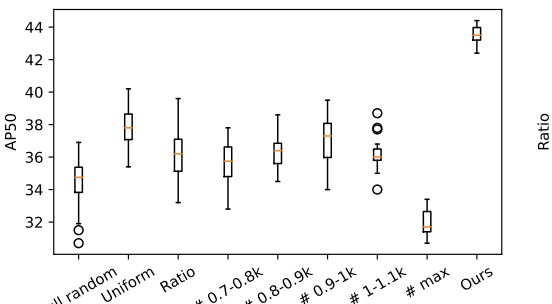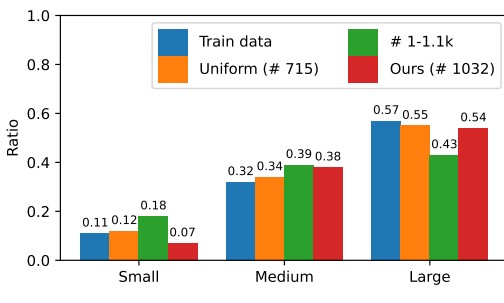

Figure 3: Left: Comparison with various selection methods. '#' denotes the number of objects in the selected data. Right: The ratio of box sizes. We followed the size criteria provided by VOC.

these cases, images already selected from one class are excluded from selection in other classes, as an image can contain objects in multiple classes. "Uniform" distributes images evenly with 10 per class, while "Ratio" selects images based on the calculated ratio of images per class.

Ours consists of 1,032 annotations. Therefore, we also experimented with random selection while controlling for the number of annotations. "# 700-1100" follows the Uniform approach but restricts the annotation count to the specified range. "# max" follows the Uniform approach but selects images based on the annotation count in descending order rather than selecting them randomly.

### 5.2.1 THE PERFORMANCE AND OBJECT SIZE RATIO.

Figure 3-Right illustrates the relationship between box size, object count, and performance. We conducted two comparisons. First, we compared the Uniform and Our selection. Then, we observed that the ratio of Uniform was closer to that of the entire dataset in terms of KL divergence, while Ours had more annotations. Second, we compared Ours and '# 1-1.1k'. Both methods had a similar number of objects, but the box size ratio of our method was closer to that of the training data. While we cannot definitively assert causality, it appears that a well-represented subset with an equal number of images has a correlation with both box size and object count.

## 5.3 ANALYSIS OF THE NUMBER OF IMAGES AND THE HYPERPARAMETER $\lambda$

### 5.3.1 THE NUMBER OF SELECTED IMAGES

Figure 4 illustrates how performance varies with the number of selected images. Since selecting 20 images indicates only one image per class is selected, $\lambda$ is meaningless. For other cases (100, 200, 500 and 1000), we set $\lambda$ as (0.0125, 0.04375, 0.04375, and 0.025), respectively. Compared to random selection (the Uniform approach), we observe that as the number of selected images increases, the performance gap naturally decreases, but it consistently remains at a high level.

### 5.3.2 BALANCE HYPERPARAMETER $\lambda$

Figure 5 illustrates the relationship between performance and $\lambda$ in Eq. 4. A high $\lambda$ value (1e+10) means selecting images based solely on cosine similarity, prioritizing representative images. In contrast, a small $\lambda$ value (1e-10) means selecting an image per class with the highest cosine similarity first and then selecting images that are as dissimilar as possible from those already selected. In other words, it emphasizes diversity from a cosine similarity viewpoint.

The observations can be made: Firstly, our approach outperforms random selection when $\lambda$ is above a certain threshold. Secondly, it is better to consider both representativeness and diversity by appropriately tuning $\lambda$ rather than simply selecting images purely based on the order of cosine similarity (1e+10). Lastly, the optimal $\lambda$ value varies depending on the number of images to be selected, as the greedy selection process (Section 4.3) progressively increases the number of selected images. Please refer to Appendix table 7 for the $AP_{50}$ values corresponding to the $\lambda$ values.

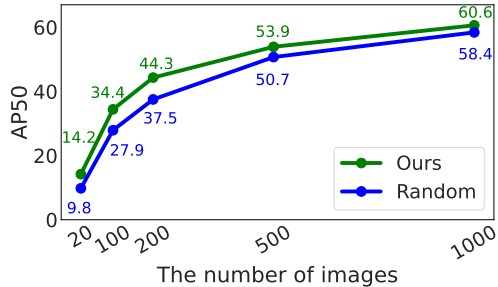
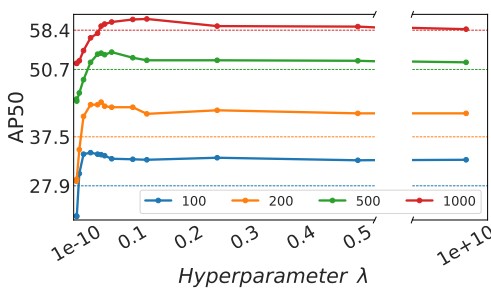

Figure 4: Performance according to the number of selected images.

Figure 5: $AP_{50}$ and $\lambda$. The dashed line represents the performance of random selection.

Table 1: The number of selected images is 200. Imagewise refers to averaging the RoI vectors and Objectwise refers to not averaging. The number of changes in the image list is based on $\lambda$=1e+10 as the reference point (The smaller $\lambda$, the severer the change). $\lambda$ is rounded to the fourth decimal place.

|  |  | The number of changes in the image list | | | | |
|---|---|---|---|---|---|---|
|  |  | 0 | 18 | 32 | 45 | 113 |
| Objectwise | $\lambda$ | 1e+10 | 0.125 | 0.075 | 0.051 | 0.015 |
|  | $AP_{50}$ | 40.4 | 40.3 | 40.7 | 41.4 | 39.0 |
| Imagewise (Ours) | $\lambda$ | 1e+10 | 0.100 | 0.050 | 0.038 | 0.013 |
|  | $AP_{50}$ | 42.1 | 43.3 | 43.5 | 43.8 | 41.5 |

## 5.4 EFFECTIVENESS OF AVERAGING THE RoI FEATURE VECTORS

### 5.4.1 PERFORMANCE COMPARISON

Table 1 presents a performance comparison between averaging the RoI feature vectors of the same class (Imagewise) or not (Objectwise). These two cases have different balance strengths for $\lambda$, as Imagewise averages within the same class, resulting in significantly fewer RoI feature vectors. Therefore, we compared the extent to which the image list changes, using 1e+10 as the reference point. Remarkably, we observed consistent high performance regardless of $\lambda$ values. However, even Objectwise outperformed random selection, achieving a result higher than 37.5 of random selection.

### 5.4.2 REPRESENTATIVENESS: OBJECTWISE VS. IMAGEWISE FEATURE VECTOR

Table 2 compares Objectwise and Imagewise based on the number of selected images when $\lambda$=1e+10. This experiment highlights that even if the cosine similarity of a single object within an image is exceptionally high, that image may not effectively represent the overall distribution of the data. For example, the case where only one image per class is selected (20 in the table) indicates how well a single image represents the corresponding class. The table shows the superiority of our imagewise selection over objectwise selection.

Table 2: $\lambda$ is 1e+10 in all cases, meaning that we selected based solely on cosine similarity ranking.

|  | 20 | 40 | 60 | 80 | 100 | 200 |
|---|---|---|---|---|---|---|
| Objectwise | 13.0 | 20.6 | 25.8 | 29.1 | 31.5 | 40.4 |
| Imagewise (Ours) | 14.2 | 23.1 | 27.3 | 30.1 | 32.9 | 42.1 |

### 5.4.3 VISULIZATION OF OBJECTWISE SELECTION

Figure 6 shows limitation examples that the objectwise approach has a possibility that the selected image may not effectively represent the entire dataset. Even if an image is selected because it contains an object with a high cosine similarity, it does not guarantee that other objects within the same image will have similarly high cosine similarities. In other words, the cosine similarity of one object in an image with all the other objects in the entire dataset may not accurately represent the cosine similarities of all objects in that image.

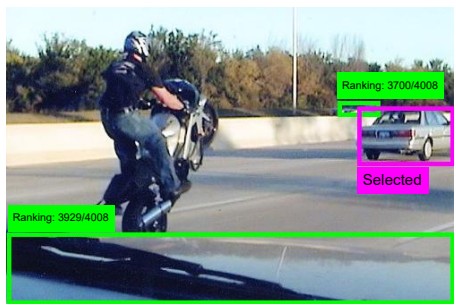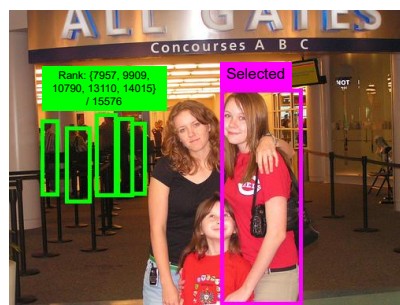

Figure 6: Examples of the Objectwise selection. Left: 'car' class. Right: 'person' class.

### 5.4.4 WHY IMAGEWISE (AVERAGING) SELECTION OVER OBJECTWISE SELECTION?

Let us compare object counts and size ratios in Section 5.2.1. In our Imagewise approach, there are 1,032 objects in our 200 selected images, which is higher than 806 in the Objectwise approach. Additionally, when considering the size ratios (small, medium, large), the Imagewise approach results in (7.3%, 38.2%, 54.5%), which is closer to the large object ratio in the entire train data, (10.5%, 32.0%, 57.5%), compared to the Objectwise, (10.2%, 37.2%, 52.6%). When we calculated the KL divergence between the distributions of the selected images and the entire training data, we found that Objectwise had a KL divergence of 0.006, lower than Imagewise's 0.013.

Table 3: Cosine similarity between the entire average features and the average feature of each image by size in 'person' class.

| Object count | 1 | 2-4 | 5 or more |
|---|---|---|---|
| Large | 0.866 | 0.931 | 0.939 |
| Medium | 0.862 | 0.915 | 0.931 |
| Small | 0.798 | 0.818 | 0.831 |

This suggests that the number of annotations played a more significant role in the performance than the size ratio in the case of Imagewise and Objectwise. Despite the higher KL divergence for Ours, there were substantial differences in the number of annotations for each size. Ours had counts of (75, 395, 562) for each size, whereas Objectwise had counts of (82, 300, 424).

We formulated a hypothesis that "As the number of objects within an image increases and their sizes are larger, the cosine similarity between the class's entire average RoI vector (class prototype) and the image's average RoI vector (image prototype) for that class will be higher."

To validate this hypothesis, we conducted the experiment presented in Table 3. Initially, we averaged all RoI vectors for the 'person' class (class prototype). Then, we made an averaged RoI vector by size within each image (imagewise-sizewise prototype). We subsequently computed the cosine similarity between the class prototype and the imagewise-sizewise prototypes. The results confirmed that the Imagewise approach leads to a higher selection of larger objects, resembling the entire dataset.

### 5.5 SAMPLING FIRST, THEN SELECTION FOR REDUCING COSINE SIMILARITY CALCULATION

Table 4 presents the experiment that we first reduced the number of data per class by random selection and then applied our method. Note that the numbers per class are not precisely equal, as some images might contain multiple classes, but rather approximations. We conducted this experiment for two reasons: handling classes with a large number of images, which can make cosine similarity calculations time-consuming, and addressing class imbalance.

Our results show a clear trend. Even with just 50 images per class, our algorithm outperformed random selection by a significant margin. Moreover, increasing the sample size to 1,000 images per class yielded similar performance. This implies that, even when dealing with classes with an unusually high number of images, as long as computational resources allow, sampling without replacement is expected to provide markedly better results than random selection. For reference, the minimum, maximum, and average number of images per class are 423, 6,469, and 1,281, respectively.

Table 4: Coreset Selection after Sampling. In all cases, we ultimately selected 200 images.

| | Random | The number of sampled image per class | | | | | | | Ours |
| | | 50 | 100 | 200 | 300 | 400 | 500 | 1000 | |
| --- | --- | --- | --- | --- | --- | --- | --- | --- | --- |
| $AP_{50}$ | 37.5 | 42.0 | 42.8 | 43.6 | 43.9 | 43.9 | 43.5 | 44.0 | 44.3 |

Table 5: COCO2017 result.

| | num img | AP | $AP_{50}$ | $AP_{75}$ |
| --- | --- | --- | --- | --- |
| Random | 400 | 6.6 | 15.1 | 4.9 |
| | 800 | 9.1 | 19.4 | 7.5 |
| Ours | 400 | 7.3 | 16.5 | 5.5 |
| | 800 | 9.6 | 20.1 | 8.1 |

Table 6: Cross architecture. We trained the models on 500 VOC images and reported $AP_{50}$.

| | RetinaNet | FCOS |
| --- | --- | --- |
| random | 54.5 | 47.9 |
| ours | 58.3 | 53.1 |
| margin | +3.8 | +5.5 |

## 5.6 EVALUATION ON THE COCO DATASET

Table 5 presents the results for COCO2017 (Lin et al., 2014), which comprises 80 classes and is considerably larger than the VOC dataset. We conducted experiments with 400 and 800 image selections. Although COCO is a larger and more challenging dataset, our approach proved its benefit, leading to improvements in AP, $AP_{50}$ and $AP_{75}$. For reproducibility, it is worth noting that the COCO2017 train set contains approximately seven times more images than the VOC07+12 trainval set. Therefore, to conduct this experiment, we initially sampled 30,000 images for each class.

## 5.7 CROSS-ARCHITECTURE EVALUATION

Table 6 presents an experiment in which we assessed whether the 500 images selected using Faster R-CNN remained effective for different networks, namely RetinaNet Lin et al. (2017) and FCOS Tian et al. (2019). We were able to confirm the effectiveness of images selected with Faster R-CNN for other networks as well. Unlike Faster R-CNN, these two networks often encountered training issues due to loss explosion when following to their respective default hyperparameters. Therefore, we adjusted the hyperparameters such as the learning rate and gradient clipping, but it is important to note that the hyperparameters for random selection and our method remained consistent. Please refer to Appendix B for reproducibility.

## 6 DISCUSSION

**Conclusion.** We have proposed a method for Coreset selection in Object Detection tasks. This method addressed the challenges specific to object detection, which involve multi-object and multi-label scenarios, different from image classification. Our approach allows for considering both representativeness and diversity while taking into account the difficulties we have outlined. Through experiments, we have demonstrated the effectiveness of our method, and its applicability to various architectures. We hope this research will further develop and find applications in diverse areas, such as dataset distillation, in the future.

**Limitation.** While our research leveraged RoI features from ground truth boxes and achieved promising results, it is important to note certain limitations. Firstly, we did not explicitly incorporate background features, which could provide additional context and potentially enhance coreset selection in object detection. Future research could explore the explicit utilization of background features. Our approach, which selects greedily on a class-by-class basis, can take into account the RoI features of the current class even when they were selected during the turn of other classes. However, our method does not simultaneously incorporate the features of other classes within the same image. Further research could explore ways to capture interactions between different classes more effectively within a single image.

**Future work.** Since our method takes localization into account, there may be aspects that can be applied to other tasks related to localization, such as 3D object detection. Furthermore, while dataset distillation has predominantly been studied in the context of image classification, it could also become a subject of research in the field of object detection datasets.

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

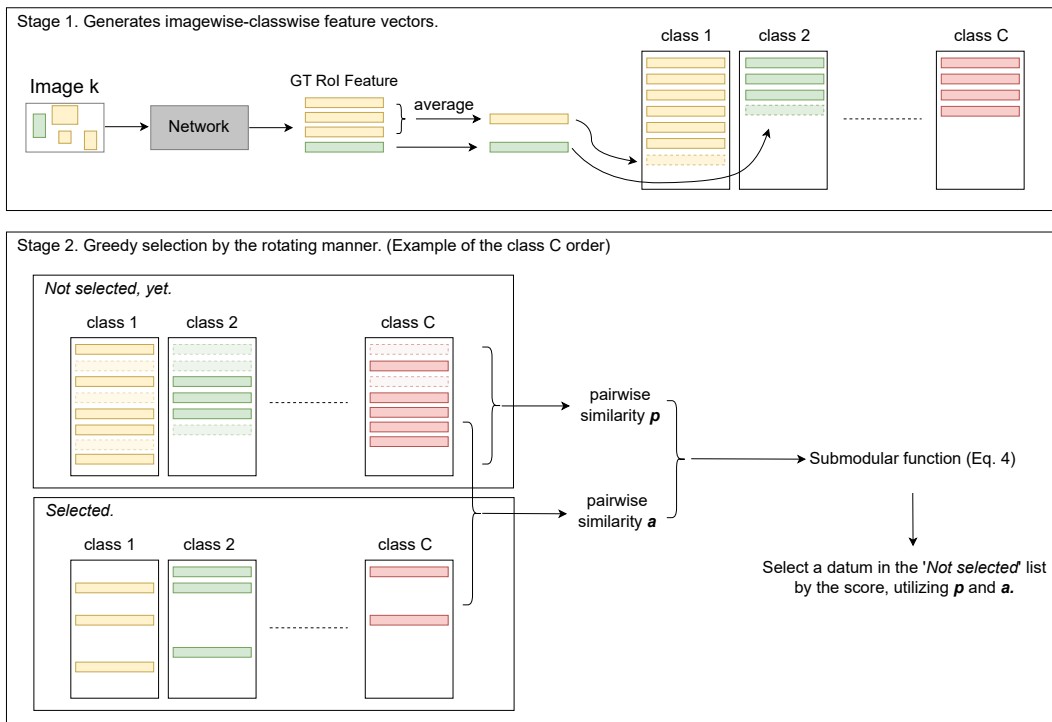

Figure 7: Stage 1 is to create imagewise-classwise feature vectors by applying RoI pooling to the ground truth boxes and subsequently averaging them by class. In Stage 2, illustrated by class C as an example, data selection for each class is performed. It calculates scores that consider the balance between previously selected and the not-selected data (Eq. 4). After selection by Eq. 4, the datum selected in the current selection step is popped from the '*Not selected, yet*' list and inserted into the '*Selected*' list. After selecting a datum for class C, the process returns to class 1 and repeats the same steps until the desired number of images is reached.

Table 7: $AP_{50}$ on Pascal VOC. The optimal $\lambda$ increases as the number of images grows.

| count | 1e-10 | 0.0005 | 0.005 | 0.0125 | 0.025 | 0.0375 | 0.04375 | 0.05 | 0.0625 | 0.1 | 0.125 | 0.25 | 0.5 | 1e+10 |
|---|---|---|---|---|---|---|---|---|---|---|---|---|---|---|
| 100 | 22.0 | 21.9 | 30.3 | 34.1 | 34.4 | 34.1 | 34.0 | 33.8 | 33.2 | 33.1 | 33.0 | 33.4 | 32.9 | 33.0 |
| 200 | 28.8 | 29.2 | 35.0 | 41.5 | 43.8 | 43.8 | 44.3 | 43.5 | 43.3 | 43.3 | 42.0 | 42.7 | 42.1 | 42.1 |
| 500 | 44.9 | 44.5 | 46.1 | 48.7 | 52.1 | 53.7 | 53.9 | 53.6 | 54.1 | 53.0 | 52.5 | 52.5 | 52.4 | 52.1 |
| 1000 | 51.9 | 51.9 | 52.4 | 54.4 | 56.9 | 57.8 | 59.2 | 59.6 | 60.0 | 60.5 | 60.6 | 59.2 | 59.1 | 58.6 |

## A ADDITIONAL IMPLEMENTATION DETAILS

Section 5.1 describes implementation details, and here we introduce additional implementation details. During training with the selected data, we used the SGD optimizer with a learning rate of 0.02, weight decay of 0.0001, and momentum of 0.9. The number of iterations was as follows: when the number of images was 200 or fewer, we trained the initialized network for 1000 iterations. For 500 images, we trained it for 2000 iterations. With 1000 images, we trained it for 4000 iterations. When there were 200 images or fewer, we performed training on the initialized network for 1000 iterations to ensure loss convergence. The learning rate was reduced to 0.1 times the initial value at 80% of the training iterations. The warm-up was conducted for 100 iterations, and there was no gradient clipping.

Regarding image sizes, we resized the shorter side to 800 pixels and maintained the original aspect ratio, ensuring the longer side remained below 1333 pixels. During training, resizing was done within the range of (480-800) with a step size of 32.

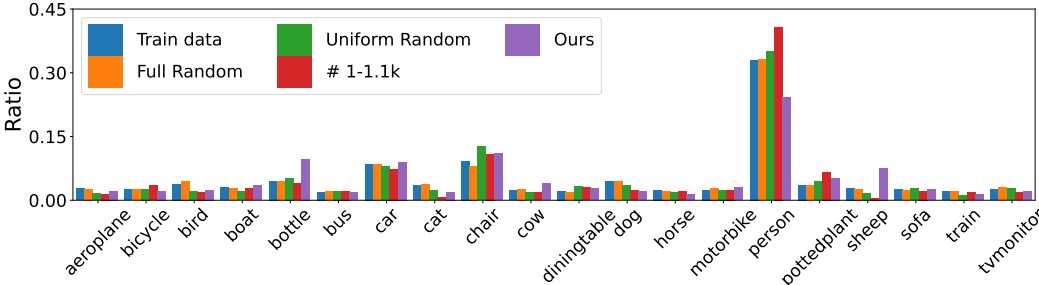

Figure 8: Annotation ratios for each class. When considering the KL divergence as the criterion with train data, Full Random was the closest. However, when calculating the balance level by computing the entropy, Ours showed the most balance.

## B    CROSS ARCHITECTURE DETAIL

In the case of RetinaNet and FCOS, as mentioned in Section 5.7, following the configurations of each detector or the configurations we experimented with in Faster R-CNN resulted in significant training instability due to loss explosion. Therefore, when training with 500 images, we followed these hyperparameters: We increased the number of training iterations from 2000 to 6000, matching the learning rate decay point at 5200 iterations accordingly. The learning rate for RetinaNet was set to 0.01, following Detectron2, while for FCOS, it was reduced from 0.01 to 0.005. We extended the warm-up iteration from 100 to 1000 iterations. Gradient clipping was introduced with a threshold of 1.0, which was previously absent. We reduced the image size in training and testing, scaling down the shorter side from 800 to 600 pixels and the longer side from 1333 to 1000 pixels. We also reduced the data resize augmentation range from 480-800 to 360-600. For other network hyperparameters, we followed Detectron2's settings for RetinaNet and used the official code's configurations for FCOS.

## C    ANALYSIS OF THE RATIO OF CLASSWISE ANNOTATION COUNTS

We explored how selection methods affect the class balance of annotations. Figure 8 shows the results. Note that in this analysis, we considered the proportions without considering the number of annotations or object sizes.

When we calculated the KL divergence with the train data, Full Random appeared to be the closest. We also evaluated how balanced the class ratios were based on the entropy. Our method showed the best balance in terms of the entropy.

Intuitively, we might expect the performance to be better when the subset's class ratios are closest to those in the train data. However, this was not necessarily the case, suggesting two possibilities. First, the number of annotations in the subsets is significantly smaller than in the train data. Therefore, having a relatively balanced dataset, even if it differs from the train data's class ratios, could be more beneficial for learning. Second, as shown in Section 5.2.1, the size of objects or the number of annotations might significantly impact performance.

## D    CORESET SELECTION WITH GRADIENTS INSTEAD OF RoI FEATURES

In our pursuit of selecting an informative subset for training, we conducted an experimental comparison where we explored the use of gradient vectors derived from RoI feature vectors instead of directly employing the RoI feature vectors. This approach was motivated by Guo et al. (2022), where they demonstrated the effectiveness of gradients in image classification tasks.

The gradient vectors were obtained by backpropagating from the classification loss. Furthermore, we performed our method with the gradient of RoI feature vectors instead of RoI feature vectors. We focused solely on the gradients of classification loss because they have significantly higher dimensions than RoI feature vectors, mainly due to the multiplication of the number of classes.

In our experiments, when selecting 200 images with $\lambda$ set to 0.05, we compared the $AP_{50}$ values. Results revealed that utilizing our method with RoI feature vectors achieved an $AP_{50}$ of 43.5, whereas when using gradients, the $AP_{50}$ was 42.3.

# E   $\lambda$ SET DIFFERENTLY FOR EACH CLASS.

We experimented with setting $\lambda$ of Eq. 4 differently for each class. Because the number of objects varies among classes, it leads to varying scales in the former term of the right-hand side of Eq. 4.

So, if we denote the number of objects in class $c$ as $N_c$, we conducted hyperparameter tuning for $\lambda_c$, such as $\lambda_c \propto 1/N_c$, $\lambda_c \propto 1/\log_2(N_c)$, or $\lambda_c \propto 1/a^{N_c}$, where $a$ is a hyperparameter. However, we observed no significant difference in $AP_{50}$. Furthermore, class-specific $AP_{50}$ values did not reveal any consistent trends concerning the number of objects.

There are several possible explanations to consider. First, our method selects classes independently, without considering other classes. However, as mentioned in Section 1, a single image can contain multiple classes. Therefore, the assumption of complete independence among classes may not hold, and $\lambda_c$ may have indirectly influence other classes. Second, the VOC dataset exhibits a relatively less severe class imbalance compared to other datasets Oksuz et al. (2020). Based on our observations that there is not a significant performance difference within a specific range of $\lambda$ values, it is possible that the imbalance is not severe.

Data imbalance is a challenging yet crucial issue addressed in many domains. Coreset selection from imbalanced data is also an area that deserves deeper exploration in the future.

# F   WHAT WOULD HAPPEN IF WE DISCARDED ALL THE SELECTED IMAGES AND CHOSE AGAIN?

We observed an $AP_{50}$ of 43.5 when we set $\lambda$ to 0.05 and selected 200 images, with 10 images per class in the VOC dataset. Subsequently, we conducted an additional experiment in which we excluded the originally chosen 200 images and then selected another 200 images. In this case, the AP50 value was 42.2.

Two crucial observations emerged from these results: Firstly, the initial selection of 200 images effectively represented the entire dataset. Secondly, even when we reselected 200 images from the remaining dataset after discarding the initial selection, the performance remained significantly superior to random selection, which yielded an $AP_{50}$ of 37.5.

