# OpenReview forum: "Coreset Selection For Object Detection"
_ICLR.cc/2024/Conference — ICLR 2024 Conference Withdrawn Submission_

### Official Review · Reviewer_daqV · 2023-10-29

**Soundness:** 3 good
**Presentation:** 3 good
**Contribution:** 2 fair
**Rating:** 5
**Confidence:** 4

**Summary:**

The paper proposes a new method for selecting a small subset of images that can represent the entire dataset for object detection tasks.  The method uses a greedy algorithm to select one image per class at a time, based on a score function that considers both representativeness and diversity. The score function is based on submodular optimization, which maximizes the benefit of adding each image to the subset. The method is evaluated on the Pascal VOC dataset and compared with various random selection methods. The method outperforms random selection by a large margin in terms of AP50 when selecting only 200 images. The method also shows its effectiveness on the COCO dataset and different network architectures.

**Strengths:**

- The current manuscript delves into the task of dataset selection in object detection.
- The concept presented in this paper is easily comprehensible and well-articulated.
- The experiments conducted on the PASCAL dataset prove the superiority of the proposed method over random sampling strategies.

**Weaknesses:**

- The greedy selection method is a heuristic approach, and the superiority of the introduced method has not been analyzed theoretically.
- The experiments conducted on the PASCAL dataset are inadequate.
- In-depth research on what types of images are suitable for object detection is lacking.

**Questions:**

- The proposed method is only compared to random sampling. It is unclear why the proposed method is superior.
- The experiment results on the COCO dataset are absent.
- The conclusions drawn in the study are inadequate, as there is a lack of discussion on which image types or annotation methods are beneficial for the object detection task.

---

### Official Review · Reviewer_M63x · 2023-10-31

**Soundness:** 3 good
**Presentation:** 3 good
**Contribution:** 2 fair
**Rating:** 3
**Confidence:** 4

**Summary:**

This paper introduces Coreset Selection for Object Detection (CSOD), a method designed to select a compact yet representative subset of the complete dataset for object detection. CSOD generates image-wise and class-wise representative feature vectors for multiple objects of the same class within each image. To ensure both representativeness and diversity, this paper proposes submodular optimization, utilizing the representative vectors in this optimization process to select a subset. Experimental results reveal that CSOD surpasses random search by 6.8% on the PASCAL VOC benchmark, even when selecting only 200 images.

**Strengths:**

1. The idea is simple, and the paper is clear and easy to follow.

2. I truly appreciate the author's inclusion of Appendix Figure 7, as it significantly aids the reader in better comprehending the concept.

3. The core formula, identified as formula (4), is logical. The initial term's objective is to select the most representative from those that have not been previously chosen, whereas the latter term aims to select an option distinct from those that have already been selected.

**Weaknesses:**

1. The author appears to be the first to propose and investigate this problem, given the absence of comparisons to other methods within this context. However, the methodology lacks rigorousness. The stated objective is to select the most representative subset from a complete dataset. Yet, the paper seems to primarily focus on selecting a subset to enhance the mAP of Faster R-CNN, which is not synonymous. Defining representativity is challenging, as it can vary across different datasets and models.

2. The results presented are not entirely persuasive. Firstly, the author does not compare the work to any other studies, but merely to a few basic counterparts such as random sampling and classwise random sampling. Even if there are no other methods available for object detection, it would be beneficial for the author to compare some classification methods and apply them to object detection.
Secondly, the paper's main experiments are based on PASCAL VOC, an older and relatively small object detection dataset. This is not particularly convincing as the resources required to train the model on the entire PASCAL VOC are minimal, rendering the selection of a small representative subset from it somewhat redundant.
Additionally, as demonstrated in Table 5, the improvement in performance on the COCO dataset is marginal when compared to the random sampling counterpart.

3. The proposed method is succinct yet lacks effectiveness. The paper's primary contribution appears to be exclusively formula (4). As demonstrated in Table 5, its performance modestly surpasses that of the random sampling counter.

**Questions:**

I have put some of my questions and doubts in the weakness sections, here are some more questions:

1. The paper only tests Faster R-CNN on COCO, how about other models?

2. How about selecting more images on COCO dataset, for example, 4000/8000? 400/800 seems meaningless given today's computation resource

---

### Official Review · Reviewer_k61v · 2023-10-31

**Soundness:** 4 excellent
**Presentation:** 4 excellent
**Contribution:** 2 fair
**Rating:** 6
**Confidence:** 4

**Summary:**

This paper proposes a method for selecting a representative subset of image data such that training on the subset provides accuracy near to training on the full dataset. Whilst this has been done with whole-image classification, the paper tackles the harder problem of object detection + classification. Results show that the approach can improve upon various random subset selection strategies, and the authors perform a fairly comprehensive analysis covering different datasets and different algorithms.

**Strengths:**

- The paper is well written, explained clearly and I would have absolutely no problem replicating their approach - a pleasure to read!

- The analysis is quite comprehensive, with the authors spending good time on showing how their approach changes. However, I don't see any values indicating the 'upper bound' of Faster-CNN trained on the full dataset?

- In brief, the approach isn't revolutionary and the results are modest, but if I had to cut down my object detection dataset I would absolutely apply this approach to give me the best chance of good performance.

**Weaknesses:**

- Results aren't as good for alternative detectors like RetinaNet and FCOS. Also, the authors mention YOLO and DETR but don't try them, not quite sure why?

- The approach's effectiveness also isn't quite as good on the harder COCO dataset (< 1% in absolute terms), though in relative terms it isn't too much different. To their credit

- Comparison against random selection is kind of vs "lowest-denominator"; it would have been nice if the authors could have used some kind of principled alternative, even one as not-for-purpose as using image classifier subset selectors such as they reference. Sure, they aren't meant for object classification, and I'd sort of expect it to do worse, but it would be a useful in placing the approach in context.

**Questions:**

- I don't see any values indicating the 'upper bound' of Faster-CNN trained on the full dataset? Why is this missing, or did I just miss it?

- What is the time taken to perform the subset extraction vs detector training time? If it is a significant fraction of training with the full dataset, then the subset extraction is arguably not helping.

---

### Official Review · Reviewer_Kdn6 · 2023-11-01

**Soundness:** 2 fair
**Presentation:** 2 fair
**Contribution:** 2 fair
**Rating:** 5
**Confidence:** 4

**Summary:**

This paper proposes a new method to select representative sets in object detection. This method averages the feature vectors of each category in the image to obtain the corresponding category prototype, and then uses a greedy algorithm to select samples. The proposed method is able to consider both the representativeness and diversity of the selected collection. Experiments show that the proposed method is significantly better than random selection on different data sets..

**Strengths:**

- This paper proposes a new method for constructing coreset in object detection. This work should be the first research to design coreset selection specifically for object detection.
- The author conducted a large number of experiments to verify the effectiveness of the method.
- The presentation is clear and easy to follow.

**Weaknesses:**

- The essence of this method is still the Core-set method, which is already very common in the field of active learning. This method may be regarded as an incremental supplement to the Core-set approach.
- The authors only compared their method with random selection. To be more convincing, comparions should be conducted with some mainstream active learning and representative selection algorithms.
- The introduction to related work is vague. The authors introduced a large number of irrelevant work such as dataset distillation. This will cause confusion to readers. At least, it should be stated in detail how these methods are related to the proposed method.

**Questions:**

- Section 3, problem setup, has only one paragraphs. It seems more reasonable to merge it with Section 4.
- The pseudo code of Algorithm 1 is not clear enough. When reading this code, the reader does not know what the variables inside mean.
- In the article, c and k are both used to represent “class”, and it is best to use unified symbols.
- There are description problems with the formula and its explanation in Section 4.3. Although the reader may guess what the authors mean, this description needs to be improved.
- The experimental setting is very weird. Since the representative samples are selected in the VOC07+12 dataset, why the model trained on the VOC07+12 dataset is used for selection. This seems unreasonable.
- Section 5.3.2, the second paragraph "The observations can be made: Firstly, our approach outperforms random selection when λ is above a certain threshold.". How was this conclusion reached? There is no relevant information in Figure 5.
- The experimental details in Section 5.4 need to be improved. In addition, how the “Objectwise” method selects samples?